# Exploring the Antioxidant Mechanisms of Nanoceria in Protecting HT22 Cells from Oxidative Stress

**DOI:** 10.3390/ijms252413281

**Published:** 2024-12-11

**Authors:** Da-Long Dong, Guang-Zhen Jin

**Affiliations:** 1Institute of Tissue Regeneration Engineering (ITREN), Dankook University, Cheonan 31116, Republic of Korea; dongdalong@dankook.ac.kr; 2Department of Nanobiomedical Science and BK21 PLUS NBM Global Research Center for Regenerative Medicine, Dankook University, Cheonan 31116, Republic of Korea; 3Department of Biomaterials Science, College of Dentistry, Dankook University, Cheonan 31116, Republic of Korea

**Keywords:** nanoceria, oxidative stress, reactive oxygen species, neuroprotection, p38 MAPK

## Abstract

An excess of reactive oxygen species (ROS), leading to oxidative stress, is a major factor in aging. Antioxidant therapies are considered crucial for delaying aging. Nanoceria, a nanozyme with antioxidant activity, holds significant potential in protecting cells from oxidative stress-induced damage. This research examines the neuroprotective role of nanoceria on HT22 cells subjected to oxidative stress induced by hydrogen peroxide (H_2_O_2_) and explores the associated molecular mechanisms. Our findings indicate that nanoceria enhances bcl-2 expression and significantly reduces Bax expression, resulting in an increased bcl-2/Bax ratio, which confirms its anti-apoptotic effect. Nanoceria boosts catalase expression and suppresses the p38 MAPK signaling pathway, indicating its role in shielding HT22 cells from oxidative stress damage induced by H_2_O_2_ through various protective mechanisms. These findings provide crucial experimental evidence for the potential applications of nanoceria in skin anti-aging and the prevention and treatment of other oxidative stress-related diseases.

## 1. Introduction

The skin, as the largest organ of the human body, is densely populated with nerve endings, making it highly sensitive to various external stimuli, including physical, chemical, and biological factors [1,2,3]. Sunburn is an acute inflammatory reaction caused by excessive exposure to ultraviolet (UV) radiation, also referred to as an oxidative stress response [4,5]. Its primary symptoms include redness, swelling, heat, and pain in the affected areas, causing significant discomfort for patients. Oxidative stress is primarily caused by the overproduction of reactive oxygen species (ROS) due to various harmful external stimuli [6,7,8]. These ROS not only directly damage skin cells but also affect other cell types throughout the body [9]. The oxidative stress damage to skin cells further stimulates the release of various inflammatory mediators from skin tissues, which, in turn, exacerbate ROS production, creating a vicious cycle that intensifies the oxidative stress state [10]. If this cycle is not interrupted in time, it may progress to chronic inflammation, severely impairing the skin’s regenerative capacity, leading to non-healing ulcers and potentially life-threatening risks [11,12,13,14,15,16,17].

Currently, most antioxidants used in antioxidant research are derived from traditional herbal medicines and various foods (such as fruits, vegetables, etc.), which contain different types of molecules with antioxidant activity [18,19,20]. The concept of nanozymes was first proposed in 2007, following the discovery of enzyme-like activity in Fe_3_O_4_ nanoparticles [21]. Nanozymes are a class of materials that possess both the unique properties of nanomaterials and the catalytic functions of enzyme mimics [22]. Since their discovery, nanozymes have been extensively studied and applied in fields such as chemical engineering, food production, agriculture, and environmental science. Recently, the ongoing progress in nanotechnology has made nanozymes a significant research focus in biomedicine [23,24]. Among them, the antioxidant properties of nanoceria have garnered significant attention [25]. Nanoceria is composed of cerium and oxygen atoms, with a fluorite structure that contains a significant number of oxygen vacancy defects [26]. These structural and chemical characteristics give nanoceria unique physicochemical properties. The study by Kwon et al. thoroughly demonstrated these properties. In their research, using a Parkinson’s disease animal model, nanoceria was used to eliminate ROS in brain lesions of neuronal cells, helping to restore normal function in neurons affected by oxidative stress [27,28,29]. This approach presents an effective treatment for neurodegenerative diseases [30]. The review article by Xue et al. provides a thorough examination of the antioxidant and antibacterial properties of nanoceria in wound healing [31].

To address the increasing prevalence of sunburn, sunscreen has become an essential tool for blocking UV radiation [32,33]. However, the potential for skin irritation and photoinstability associated with organic sunscreens has prompted researchers to explore inorganic alternatives. Among the commonly used inorganic sunscreens, titanium dioxide (TiO_2_) and zinc oxide (ZnO) are the key active ingredients [34,35]. However, recent studies have raised concerns about the cytotoxic effects of ZnO [36,37]. Carrillo-Romero et al. investigated the cytotoxicity of several nanomaterials, including TiO_2_, ZnO, silicon dioxide (SiO_2_), nanoceria, silver (Ag), and multi-walled carbon nanotubes (MWCNTs), using cell lines representing the human respiratory, digestive, skin, and immune systems. Their findings revealed that ZnO exhibited significant cytotoxicity in the skin, digestive, and respiratory system cell lines. Silver nanoparticles also showed considerable toxicity to digestive system cells, while SiO_2_ induced pro-inflammatory responses in immune cells. In contrast, nanoceria demonstrated anti-inflammatory properties, and TiO_2_ exhibited no cytotoxicity across all tested cell lines. The results emphasize nanoceria’s potential as a promising inorganic oxide substitute in sunscreens, offering significant advantages in terms of safety and functionality [37].

Concerns about the potential adverse effects of nanomaterials on human health are widespread. From a toxicological perspective, the toxicity of nanoceria is influenced by the content of surface Ce^3+^, particle size, and surface area, with the primary toxic mechanisms involving oxidative stress and inflammation. In studies on nanoceria-induced pulmonary fibrosis, Lin et al. found that exposure to nanoceria increased the expression of transforming growth factor-β1 (TGF-β1), and the harmful effects on the respiratory system were associated with its particle size (7–10 nm) [38]. The toxic effects of nanoceria on the nervous system are particularly concerning for children and the elderly. Research by Sethi et al. showed that damage to nerve cells induced by nanoceria is linked to the activity of nitric oxide synthase [39]. The liver plays a crucial role in mitigating the toxicity of foreign drugs and nanomaterials. You et al. indicated in their study that the hepatotoxicity of nanoceria is related to oxidative stress, but the specific mechanisms by which different shapes and sizes induce hepatotoxicity still require further investigation [40].

This study investigated the neuroprotective effects of nanoceria on hydrogen peroxide (H_2_O_2_)-induced oxidative stress in HT22 cells. We selected a neuronal cell line as an in vitro model because nerve cells are highly sensitive to acute inflammatory responses induced by UV radiation, making them an ideal system for validating the antioxidant properties of nanoceria. To comprehensively assess the biocompatibility of nanoceria, we conducted cytotoxicity screening and systematically analyzed its compatibility with cells to ensure safety and efficacy for potential applications. Superoxide dismutase (SOD), glutathione peroxidase (GPx), and catalase (CAT) are essential antioxidant enzymes in the body. SOD interacts with free radicals to produce H_2_O_2_, which is then broken down into water and oxygen by GPx and CAT [36]. To further elucidate the molecular mechanisms behind the antioxidant protection of nanoceria, we performed Western blot analysis to evaluate the expression of key apoptosis-related proteins, including the anti-apoptotic protein bcl-2, the pro-apoptotic protein Bax, and stress markers p38 and phosphorylated p38 (p-p38), and to detect the expression of endogenous CAT with antioxidant activity. The results showed that nanoceria effectively reduced ROS-induced pro-apoptotic signaling, enhanced the bcl-2/Bax ratio, upregulated CAT expression, and inhibited P38 activation. These findings indicate that nanoceria not only possesses excellent antioxidant properties but also provides protective effects by regulating key signaling pathways. This study provides strong scientific evidence for the potential application of nanoceria in protecting cells from oxidative damage and promoting tissue repair.

## 2. Results

### 2.1. The Neuroprotective Effect of Nanoceria on HT22 Cells Under H_2_O_2_-Induced Oxidative Stress

#### 2.1.1. The Impact of Varying H_2_O_2_ Concentrations on HT22 Cell Viability

Different concentrations of H_2_O_2_ (50, 100, 250, 500, or 750 µM) were used to treat HT22 cells for 30 min to evaluate their viability, followed by a survival rate assessment. The results indicated a minor, non-significant increase in cell viability at concentrations between 50 and 250 μM. Studies have indicated that moderate concentrations of H_2_O_2_ can act as a signaling molecule, promoting cell growth and proliferation [9]. This phenomenon may be related to the dual role of H_2_O_2_ in cellular signaling pathways. However, after treatment with 500 μM and 750 μM H_2_O_2_, cell viability decreased to 76.76 ± 13.77% and 68.81 ± 8.40%, respectively (Figure 1). We speculate that extremely low cell viability may hinder the accurate assessment of nanoceria’s antioxidant effects. Therefore, 500 μM H_2_O_2_ was chosen for the following experiments.

#### 2.1.2. The Impact of Varying Nanoceria Concentrations on HT22 Cell Viability

The optimal nanoceria concentration was identified by treating cells with varying doses (10, 20, 40, 80, or 160 μg/mL) for 24 h and evaluating cell viability via the CCK-8 assay. The results indicated cell viability percentages of 100.72 ± 2.23%, 103.52 ± 0.97%, 107.48 ± 2.72%, 97.49 ± 3.11%, and 94.03 ± 1.99% following treatment with varying concentrations of nanoceria. Both the 20 μg/mL and 40 μg/mL groups showed a significant increase in cell viability, with the 40 μg/mL group having the highest viability. The 80 μg/mL and 160 μg/mL groups showed reduced cell viability compared to the control, suggesting that nanoceria concentrations above 80 μg/mL may induce cytotoxic effects (Figure 2). Therefore, a treatment concentration of 40 μg/mL of nanoceria was selected for studying the antioxidant mechanisms.

The size, concentration, exposure duration, and cell type of nanoparticles are critical factors influencing cell viability. In this study, nanoceria smaller than 25 nm were used, and cell viability was evaluated 24 h after treatment. The results indicated a decreasing trend in cell viability at a concentration of 160 μg/mL. Ma et al. treated ARPE-19 cells with nanoceria of 15, 30, and 45 nm for 24 and 48 h, respectively, and assessed cell viability. Their findings showed that 15 nm of nanoceria caused a significantly greater reduction in cell viability compared to 30 and 45 nm of nanoceria [41]. Similarly, Cheng et al. exposed SMMC-7721 cells to nanoceria of 20–30 nm at concentrations of 0, 12.5, 25, 50, 100, and 200 μg/mL for 72 h. Their results demonstrated a significant decrease in cell viability when the concentration reached 50 μg/mL [42]. Moreover, the toxicological evaluation of nanoparticles in vivo has become a key area of interest for researchers. Studies have revealed that more than 90% of nanoparticles accumulate in the liver after entering the body, and hepatotoxicity is typically observed only at high doses. Tseng et al. intravenously administered 30 nm of nanoceria at a dose of 85 mg/kg to rats and monitored them for 90 days. The results showed a significant increase in hepatocyte apoptosis along with abnormal changes in liver function [43].

#### 2.1.3. The Effect of Nanoceria on the Viability of HT22 Cells Under H_2_O_2_-Induced Oxidative Stress

After treatment with 500 μM H_2_O_2_, different concentrations of nanoceria were added to assess their ability to alleviate oxidative damage. Cell viability in the H_2_O_2_-only treatment group decreased to 77.23 ± 6.21% compared to the control group. Conversely, the cell viability in groups treated with nanoceria at concentrations of 10, 20, 40, or 80 μg/mL was 122 ± 6.78%, 131.93 ± 5.16%, 129.56 ± 7.80%, and 110.21 ± 4.24%, respectively, all notably exceeding the control group’s viability. Cell viability increased most significantly in the 20 μg/mL and 40 μg/mL groups. Cell viability significantly increased in all nanoceria-treated groups compared to the H_2_O_2_-only group, with the most notable improvements at 20 μg/mL and 40 μg/mL concentrations (Figure 3).

A live/dead cell staining kit was used to assess cell viability and survival, and the results were observed with a fluorescence microscope. The results demonstrated that the control group maintained stable cell viability with no significant cell death (Figure 4A,D). In contrast, in the H_2_O_2_-treated groups, cells marked by red fluorescence, indicative of cell death, became increasingly apparent, especially in the group treated with H_2_O_2_ alone (Figure 4B,E). Notably, compared to the H_2_O_2_-only treatment group, the nanoceria-treated group showed a significant reduction in cell death rate (Figure 4C,F). These findings suggest that H_2_O_2_-induced oxidative stress is the primary cause of cell death, while nanoceria, through its free radical scavenging activity, effectively alleviated H_2_O_2_-induced cellular damage. To quantitatively assess cell viability, we conducted cell counting analysis using the ImageJ 1.52P software to quantify live and dead cells (Figure 4G). The quantitative results were in strong agreement with the morphological observations, demonstrating that nanoceria exerted significant neuroprotective effects, effectively counteracting oxidative stress damage induced by H_2_O_2_. Overall, these data provide compelling evidence that nanoceria effectively protects HT22 cells from oxidative stress-induced cellular damage.

### 2.2. The Antioxidant Mechanism of Nanoceria

#### 2.2.1. The Impact of Nanoceria on Apoptosis-Related Protein Expression Triggered by H_2_O_2_

To assess the impact of oxidative stress and antioxidants on the regulation of cell apoptosis, the expression levels of apoptosis-related proteins bcl-2 and Bax under different treatment conditions are illustrated in Figure 5. Bcl-2 functions as an anti-apoptotic protein, whereas Bax serves as a pro-apoptotic protein. The apoptotic propensity of cells is often evaluated using the bcl-2/Bax ratio. Figure 5A illustrates the variation in bcl-2 and Bax protein expression across the control, H_2_O_2_ treatment, and nanoceria treatment groups. β-actin, used as a housekeeping protein, was used to standardize the expression levels of bcl-2 and Bax. The quantitative analysis indicates that the bcl-2/β-actin ratio (Figure 5B) is significantly elevated in the nanoceria treatment group compared to both the control and H_2_O_2_ groups (*p* < 0.05). Additionally, bcl-2 expression is notably reduced in the H_2_O_2_ group relative to the control group (*p* < 0.01). This indicates that H_2_O_2_ treatment could increase apoptosis by reducing the expression of the anti-apoptotic protein bcl-2. In contrast, the increased bcl-2 expression in the nanoceria group suggests its potential anti-apoptotic effect. The Bax/β-actin ratio (Figure 5C) reveals a statistically significant increase in Bax expression in the H_2_O_2_ treatment group compared to the control and nanoceria groups, suggesting that H_2_O_2_ treatment enhances apoptotic pathway activation. Bax expression in the nanoceria group is significantly reduced compared to the H_2_O_2_ group, nearing control levels (*p* < 0.05), indicating that nanoceria may inhibit apoptosis by decreasing Bax expression. The bcl-2/Bax ratio is significantly higher in both the control and nanoceria groups compared to the H_2_O_2_ group (*p* < 0.01) as shown in Figure 5D. This further supports the potential role of nanoceria treatment in inhibiting cell apoptosis. In conclusion, the findings suggest that H_2_O_2_ treatment induces apoptosis by decreasing bcl-2 and increasing Bax expression, whereas nanoceria treatment demonstrates notable anti-apoptotic effects, possibly by enhancing bcl-2 and suppressing Bax expression. These data provide important experimental evidence for the potential applications of nanoceria in antioxidation and cell protection.

#### 2.2.2. The Effect of Nanoceria on Cell Catalase Expression Under Oxidative Stress Conditions

Figure 6 illustrates catalase expression levels, an antioxidant enzyme, across various treatments to assess oxidative stress and antioxidant impacts on the cellular antioxidant system. Western blot analysis (Figure 6A) revealed differences in catalase expression levels among the control group, H_2_O_2_ treatment group, and nanoceria treatment group. β-actin was used as a housekeeping protein to normalize the catalase expression levels. The catalase/β-actin ratio (Figure 6B) was employed in the quantitative analysis to assess variations in catalase expression among the treatment groups. The results indicated that the H_2_O_2_ treatment group had increased catalase levels compared to the control group, suggesting that cells may enhance their antioxidant capacity by increasing catalase expression in response to oxidative stress. The nanoceria treatment group exhibited significantly higher catalase expression levels compared to the H_2_O_2_ and control groups, suggesting that nanoceria enhances cellular antioxidant defense by upregulating catalase expression due to its strong catalytic antioxidant properties. Overall, these results suggest that H_2_O_2_ treatment induces catalase expression to mitigate oxidative stress, while nanoceria treatment results in a more pronounced upregulation of catalase, supporting its potential role in enhancing cellular antioxidant capacity. These findings provide experimental evidence to better understand the regulatory mechanisms of the cellular antioxidant enzyme system under different treatment conditions.

#### 2.2.3. By Adjusting the Phosphorylation Level of p38 MAPK, Nanoceria Offers Protection to Neurons from Oxidative Stress

Figure 7 shows the activation of the p38 MAPK signaling pathway in response to oxidative stress and the impact of nanoceria intervention. Figure 7A presents Western blot data illustrating the expression levels of phosphorylated p38 (p-p38) and total p38 protein across the control, H_2_O_2_ treatment, and nanoceria treatment groups. According to quantitative analysis, H_2_O_2_ treatment leads to a significant rise in p-p38 expression levels relative to the control group (*p* < 0.01). In contrast, nanoceria significantly inhibits H_2_O_2_-induced p38 phosphorylation (*p* < 0.01), reducing it to levels close to the control group (Figure 7B). Notably, there are no significant differences in total p38 protein expression levels among the three groups (Figure 7C), indicating that the treatment conditions mainly affect p38 phosphorylation rather than total protein expression. The p-p38 to total p38 ratio (Figure 7D) is significantly elevated in the H_2_O_2_ treatment group compared to the control group (*p* < 0.01), while the nanoceria treatment group shows a significantly reduced ratio compared to the H_2_O_2_ group (*p* < 0.01). This confirms nanoceria’s regulatory influence on the p38 MAPK signaling pathway. The study indicates that oxidative stress triggers the p38 MAPK pathway to manage cellular stress, and nanoceria may offer neuroprotection by preventing excessive p38 phosphorylation. This finding indicates that oxidative stress-induced cellular damage may involve the p38 MAPK signaling pathway and identifies a new molecular target for exploring nanoceria’s antioxidative protective mechanisms. Additionally, as a critical regulator of stress responses, changes in the phosphorylation level of p38 MAPK may be associated with the expression and activation of multiple downstream effector molecules, offering important insights for further investigation into the molecular mechanisms of oxidative stress damage.

## 3. Discussion

Through systematic screening and optimization experiments, this study comprehensively explored the multi-level protective mechanisms of nanoceria in resisting oxidative stress in HT22 cells. First, a stable and reliable H_2_O_2_-induced oxidative stress model was established, and 500 μM H_2_O_2_ was identified as the optimal concentration, which caused significant cell damage while ensuring sufficient cell survival rate. This provided a solid experimental foundation for the subsequent study of the protective mechanisms of nanoceria. In this study, we confirmed that nanoceria not only regulates apoptosis factors but also activates the expression of endogenous antioxidant enzymes. With its powerful antioxidant properties, nanoceria fully exerted its multi-level protective effect on HT22 cells under oxidative stress conditions.

Nanoceria not only mimics the activity of SOD by reacting with free radicals to generate H_2_O_2_ but also demonstrates CAT-like (CAT activity by decomposing H_2_O_2_ into water and oxygen. This remarkable characteristic of nanoceria is attributed to the dynamic and reversible transitions between its two oxidation states, Ce^3+^ and Ce^4+^ [35,44,45,46]. The corresponding catalytic reactions are as follows:(1)$$O_{2}^{•-}+{\mathrm{Ce}}^{3+}+2H^{+}\to H_{2}O_{2}+{\mathrm{Ce}}^{4+}$$
3H_2_O_2_ + 2Ce^4+^ → 2H_2_O + 2O_2_ + 2H^+^ + 2Ce^3+^(2)

From the above catalytic reaction equations and Figure 8, we can see how nanoceria mimics the dual enzyme functions of SOD and CAT, thereby confirming its exceptional antioxidant capacity.

Nanoceria significantly influences cell apoptosis by upregulating the anti-apoptotic protein bcl-2, downregulating the pro-apoptotic protein Bax, and notably increasing the bcl-2/Bax ratio. Furthermore, another important finding of this study is that nanoceria significantly enhances the expression levels of the endogenous antioxidant enzyme catalase. Although the exact mechanism remains to be further explored, one possible explanation is that nanoceria facilitates the enzymatic breakdown of H_2_O_2_, thereby preserving a portion of catalase to bolster the cell’s defense capabilities against future oxidative stress. This study also reveals that nanoceria inhibits the p38 MAPK signaling pathway. The phosphorylation of p38 MAPK is linked to the expression and activation of downstream effectors, including the NF-κB signaling pathway, which is crucial in inflammatory stress responses and may worsen oxidative stress-induced cellular damage [48]. By inhibiting the phosphorylation of p38 MAPK, nanoceria significantly reduces the activation of downstream stress response signaling molecules, thereby breaking the vicious cycle of oxidative and inflammatory stress-induced damage. This discovery offers a novel perspective for addressing oxidative damage and provides an important foundation for developing potential therapeutic strategies.

The study by Amin et al. demonstrated that intraperitoneal injection of 25 nm of nanoceria (at a dose of 0.01 μg/kg body weight) significantly enhanced the synthesis of SOD and CAT, thereby providing protective effects against monocrotaline-induced liver injury in rats [49]. Similarly, an in vivo animal study conducted by Hashem et al. revealed that in a rat model of liver injury induced by d-galactosamine and lipopolysaccharide, intraperitoneal administration of the same dose of nanoceria significantly promoted the synthesis of SOD, GPx, and CAT, thereby exhibiting protective effects against liver damage [50]. In our in vitro study, nanoceria treatment increased the expression of CAT, one of the key antioxidant enzymes. We speculate that this treatment may also enhance the synthesis of other antioxidant enzymes, such as SOD and GPx. MAPKs consist of three subfamilies—p38, ERK1/2, and JNK—that are widely involved in various physiological processes. Among them, p38 and JNK are closely associated with inflammatory responses induced by pro-inflammatory factors, whereas ERK1/2 MAPK is primarily linked to cell division and proliferation [51]. Zhao et al. investigated the neuroprotective effects of edaravone under oxidative stress conditions in neuronal cells. Using hydrogen peroxide to induce oxidative stress in HT22 cells, their study demonstrated that the three MAPK subfamilies (ERK1/2, JNK, and p38 MAPK) are closely associated with oxidative stress. Further analysis confirmed that edaravone exerts neuroprotective effects by blocking these signaling pathways [52]. The antioxidant effects of nanoceria are achieved not only by promoting the synthesis of endogenous antioxidant enzymes and inhibiting the MAPK signaling pathways but also by enhancing antioxidant capacity through the activation of the PI3K/Akt/mTOR signaling pathway [40]. Metformin, a well-known drug widely used to treat type II diabetes, has also shown antioxidant properties. Khallaghi et al. demonstrated that treating PC12 cells with metformin under hydrogen peroxide-induced oxidative stress significantly increased cell survival rates. The treatment promoted the synthesis of SOD, GPx, and CAT, as well as activated the PI3K/Akt/mTOR signaling pathway, effectively rescuing cells from oxidative damage [53]. Based on these findings, it is necessary to further investigate the impact of the PI3K/Akt/mTOR signaling pathway on cell survival in future studies to provide more robust and reliable experimental evidence for our research hypothesis.

In this study, we present comprehensive experimental data on the antioxidant stress properties of nanoceria. To consider nanoceria as a potential replacement for currently used inorganic oxides in inorganic sunscreens, it is necessary to further examine aspects such as UV protection, photostability, and oxidative stability. Caputo et al. demonstrated that nanoceria not only possesses strong antioxidant properties but also effectively shields against UV radiation, thereby protecting cells from UV-induced damage [54]. Zholobak et al. conducted detailed studies on the UV shielding performance and photocatalytic activity of nanoceria, showing that nanoceria’s UV shielding is comparable to that of TiO_2_ and ZnO nanoparticles. Additionally, they found that nanoceria exhibits relatively low photocatalytic activity, which decreases as nanoparticle size decreases [55]. Oxidative stability is a key factor in evaluating the long-term environmental use of nanomaterials. Cerium oxide, with its stable fluorite-type crystal structure and reversible Ce^3+^/Ce^4+^ redox properties, can maintain stability by absorbing or releasing oxygen. This feature enables it to retain excellent chemical stability under high-temperature and harsh oxidative conditions [56,57].

The interactions of nanoceria in biological and non-biological environments are intricately tied to its physicochemical properties and reactivity. These properties can vary significantly depending on factors such as particle size, shape, surface charge, and surface coatings. The distinctive characteristics of nanoceria are largely influenced by its dual oxidation states, particularly the interconversion between Ce^3+^ and Ce^4+^, which exhibit both oxidative and reductive behaviors. Additionally, surface adsorption phenomena driven by hydroxyl groups further shape its properties [58,59]. Studies have revealed that the properties of nanoceria can undergo substantial changes under varying environmental conditions, including particle aging, storage methods, concentration, and environmental factors such as pH, ionic strength, and the presence of redox-active substances [60]. A thorough evaluation of the dynamic transformations of nanoceria in different environments is therefore essential for ensuring its safe and effective application. In physiological environments, the strong oxidative properties of nanoceria may lead to the oxidation of monoamine neurotransmitters and the natural antioxidant vitamin C, potentially generating toxic metabolites and resulting in neurotoxic effects [61,62,63]. Gunawan et al. demonstrated that proteins at high concentrations readily adhere to the surface of nanoceria particles, forming a protein corona that influences their transport characteristics, accumulation, and uptake in both in vivo and in vitro systems [64]. Furthermore, research by Patil et al. revealed that positively charged nanoceria adsorbs significantly more bovine serum albumin compared to negatively charged particles, which exhibit little to no protein adsorption. This differential adsorption behavior contributes to a preferential cellular uptake of positively charged particles [65]. Consequently, when assessing the antioxidant activity or cytotoxicity of nanoceria in ion-rich biological environments, it is crucial to account for the potential influence of surface adsorption on experimental outcomes.

The biological distribution and final fate of nanomaterials within the body are of significant importance from a health-maintenance perspective. Nanoparticles, including cerium oxide nanoparticles, are primarily distributed to the liver and spleen (approximately 90–95%) following intravenous injection, with about 9% found in the kidneys, and minimal or undetectable levels in the lungs and other organs. At prescribed therapeutic doses, cerium oxide does not induce toxicity, even when it remains in the liver for several months. During this period, cerium oxide degrades into harmless Ce^3+^ ions and is excreted through the kidneys. However, high doses may result in significant toxic reactions [66]. Yokel et al. observed the distribution of cerium oxide nanoparticles of different sizes in rats after 30 days of intravenous injection and found that smaller nanoparticles (5 nm) tended to accumulate in the liver, whereas larger nanoparticles (15 nm and 30 nm) were more likely to accumulate in the spleen [67]. Hirst et al. administered 3–5 nm of cerium oxide nanoparticles via oral, intravenous, and intraperitoneal routes and discovered that the liver and spleen had the highest distribution levels following intravenous and intraperitoneal injection, while the oral administration exhibited the least distribution [68]. Konduru et al. found that after administering cerium oxide nanoparticles through the airway, stomach, and intravenous routes, the intravenous administration showed the most significant distribution in the liver. Oral administration resulted in low gastrointestinal absorption, with most of the particles being excreted through feces. The airway administration showed the least distribution outside the lungs [69]. Heckman et al.’s research indicated that, in the absence of toxicity, cerium oxide could be excreted through feces and urine after biological processing or dissolution [70]. Muhammad et al. reported that in the liver, cerium oxide nanoparticles slowly dissolve and undergo biotransformation in physiological media, gradually reducing in size over time until its complete degradation [71].

## 4. Conclusions

Currently, zinc oxide, a widely used component in commercial physical sunscreens, has raised concerns due to the increasing amount of experimental data indicating its cytotoxicity. This highlights the urgent need for safer and more effective alternatives. In contrast, research on the safety and immune-enhancing properties of cerium oxide nanoparticles (nanoceria) has been steadily growing. In this study, we systematically provide comprehensive experimental data and molecular evidence on the antioxidant properties of nanoceria, revealing its multi-layered protective mechanisms against oxidative stress. Combined with literature reports on the excellent UV-shielding, photostability, and oxidative stability of nanoceria, our findings offer strong scientific evidence supporting its potential as a cellular antioxidant protector and tissue repair material in the skincare industry and related medical applications.

## 5. Materials and Methods

### 5.1. Cell Culture

The HT22 cell line, derived from mouse hippocampal cells, was acquired from the American Type Culture Collection. The culture medium for the cells was high-glucose DMEM, supplemented with 10% FBS and 1% penicillin/streptomycin (100 U/mL penicillin, 100 µg/mL streptomycin; Sigma Aldrich, St. Louis, MO, USA). At 80% confluence, cells were either subcultured, used for experiments, or cryopreserved.

### 5.2. Analysis of Cell Viability Under Varying Concentrations of H_2_O_2_

HT22 cells were plated at 5000 cells per well, with H_2_O_2_ concentrations of 50, 100, 250, 500, and 750 µM. The CCK-8 assay, provided by Dojindo Molecular Technologies (Kumamoto, Japan), was used to evaluate cell viability following a 24 h incubation period. To each well, 100 µL of CCK-8 solution diluted to 1:10 was added. The plate was incubated at 37 °C for 2 h, and absorbance was read at 450 nm with a microplate reader.

### 5.3. Analysis of Cell Viability Under Varying Concentrations of Nanoceria

Nanoceria (CeO_2_ nanoparticles) was purchased from (Cerium (IV) oxide, <25 nm, product number: 544841; Sigma-Aldrich, St. Louis, MO, USA). In 96-well plates, cells were seeded at a concentration of 5000 per well. Cells were exposed to nanoceria at concentrations of 10, 20, 40, 80, and 160 µg/mL after 24 h. Following a 24 h exposure period to nanoceria, the cell viability analysis was conducted as described earlier. Using a microplate reader, absorbance at 450nm was determined following a 2 h incubation period with the CCK-8 reagent at 37 °C.

### 5.4. Analysis of Antioxidant Response of Nanoceria

After H_2_O_2_ treatment, the neuroprotective effect of nanoceria was analyzed using CCK-8 cell viability assay and live/dead fluorescent staining, respectively. The experiment comprised three groups: Group1 (Control), Group2 (H_2_O_2_-treated), and Group3 (H_2_O_2_-treated with nanoceria). The analysis was conducted using a 96-well plate, with cell seeding methods as described earlier. After a 24 h culture period, Groups 2 and 3 were treated with H_2_O_2_ for 30 min, followed by PBS washes. Fresh medium was added to Groups 1 and 2, while Group 3 received fresh medium with 40 μg/mL nanoceria. HT22 cell viability was evaluated with the CCK-8 reagent 24 h post-nanoceria treatment. The detailed procedure is as described earlier. For 15 min, cells were stained with live/dead fluorescent dye at room temperature in the dark using the Live/Dead^®^ Cell Viability Assay Kit (R37601, Life Technologies, Carlsbad, CA, USA). After incubation, the cells were washed with PBS and observed using an inverted fluorescence microscope (DP2-BSW, Olympus Co., Tokyo, Japan) with a DP-72 digital camera for capturing images.

### 5.5. Western Blot

Cells for Western blotting were seeded at a density of 3 × 10^5^ per 60 mm culture dish. Treatments using 500 μM H_2_O_2_ and 40 μg/mL nanoceria were conducted following established protocols. Cells were harvested post-experiment via centrifugation at 2000 rpm for 3 min. Cells were digested with trypsin and then lysed on ice for 30 min using a lysis buffer supplemented with protease and phosphatase inhibitors (Halt™ Protease and Phosphatase Inhibitor Cocktail, 100X, Thermo Scientific, USA; EBA-78440, Elpis Biotech, Daejeon, Republic of Korea). The lysates were spun at 10,000 rpm for 10 min at a temperature of 4 °C, following the manufacturer’s guidelines, and protein concentration in the supernatant was determined using the Pierce BCA protein assay kit (Thermo Scientific). Samples were denatured by heating at 100 °C for 10 min, followed by separation via SDS-PAGE. Proteins were transferred to a PVDF membrane following electrophoresis. BSA (SolMate BSA Grade IY; GeneAll Biotechnology, Seoul, Republic of Korea) was used to block the membrane for 60 min, and it was then incubated overnight at 4 °C with primary antibodies against bcl-2, Bax, catalase (Santa Cruz Biotechnology, Dallas, TX, USA), phosphorylated p38 (p-p38), p38, and β-actin(Cell Signaling, Danvers, MA, USA). On the subsequent day, the membranes underwent three washes with TBST and were incubated with HRP-conjugated secondary antibodies for 60 min. The detection of protein signals was carried out using the LAS4000mini imaging system from Sweden, utilizing SuperSignal West Pico and PLUS chemiluminescent substrates from (Thermo Scientific, Waltham, MA, USA). The ImageJ 1.52P software was utilized for quantitative analysis. Table 1 presents the antibodies utilized in this study.

### 5.6. Statistical Analysis

The statistical analysis was performed with GraphPad Prism 8.0.2, and data are expressed as mean ± standard deviation. Statistical significance among the research groups was assessed using one-way ANOVA and post hoc Tukey’s test. The quantitative analysis of Western blots and live/dead staining images was performed using the ImageJ 1.52P-win64 software. *p*-values less than 0.05 were regarded as statistically significant.

## Figures and Tables

**Figure 1 ijms-25-13281-f001:**
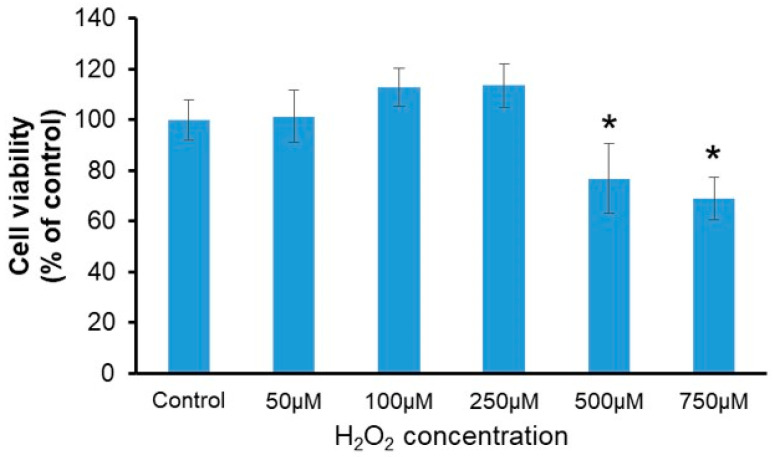
Assessment of cell viability following exposure to different H_2_O_2_ concentrations. * *p* < 0.05 compared to the control group (n = 3).

**Figure 2 ijms-25-13281-f002:**
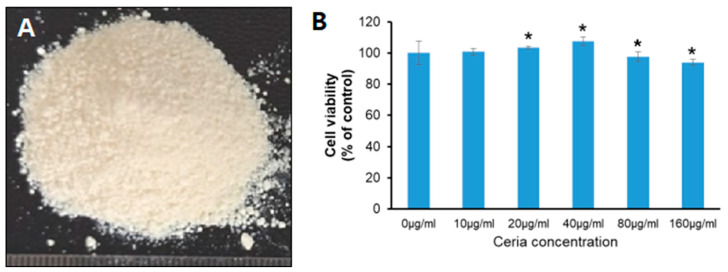
Cell viability following exposure to different nanoceria concentrations. (**A**) The morphology of nanoceria. (**B**) The impact of ceria on HT22 cell viability under H_2_O_2_-induced oxidative stress was assessed, with significance indicated by * *p* < 0.05 compared to the control group (n = 3).

**Figure 3 ijms-25-13281-f003:**
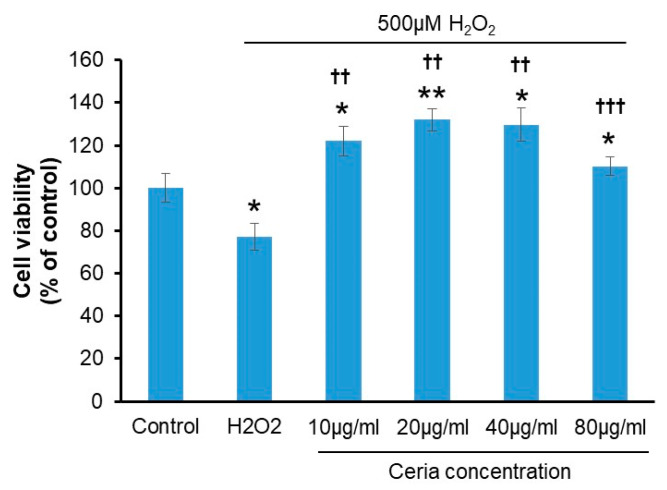
Ceria’s impact on oxidative stress induced by H_2_O_2_ in HT22 Cells. * *p* < 0.05 and ** *p* < 0.01 compared to the control group; ^††^ *p* < 0.01, and ^†††^ *p* < 0.001 compared to the H_2_O_2_-treated group (n = 3).

**Figure 4 ijms-25-13281-f004:**
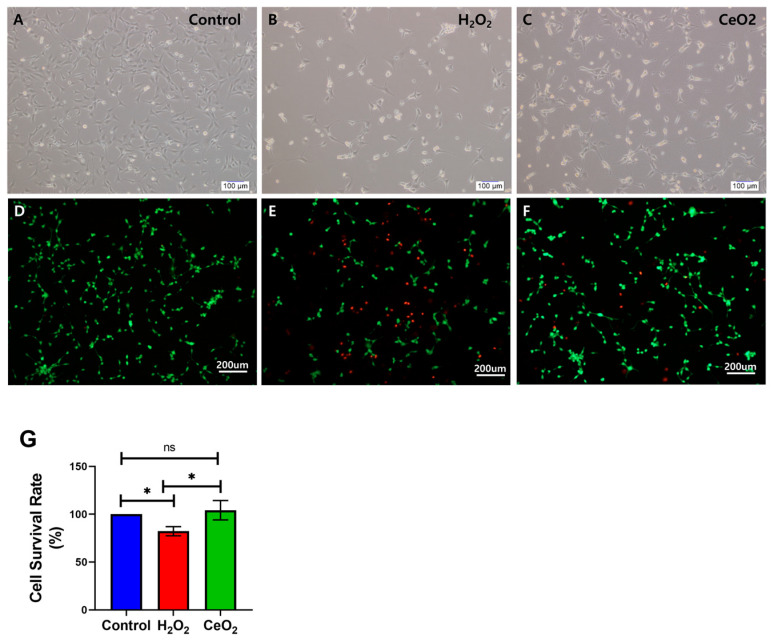
The effect of nanoceria on the viability of HT22 cells after H_2_O_2_ treatment was assessed through live/dead cell staining analysis. (**A**–**C**) Phase contrast micrograph images; (**D**–**F**) fluorescent imaging of live/dead cell staining. (**A**,**D**) Control group; (**B**,**E**) H_2_O_2_-treated group; (**C**,**F**) nanoceria-treated group. (**G**) Quantitative evaluation of HT22 cell viability using the ImageJ 1.52P software. Statistical significance was determined with * when *p* < 0.05, while results were considered not significant (ns) if *p* ≥ 0.05, based on a sample size of n = 3.

**Figure 5 ijms-25-13281-f005:**
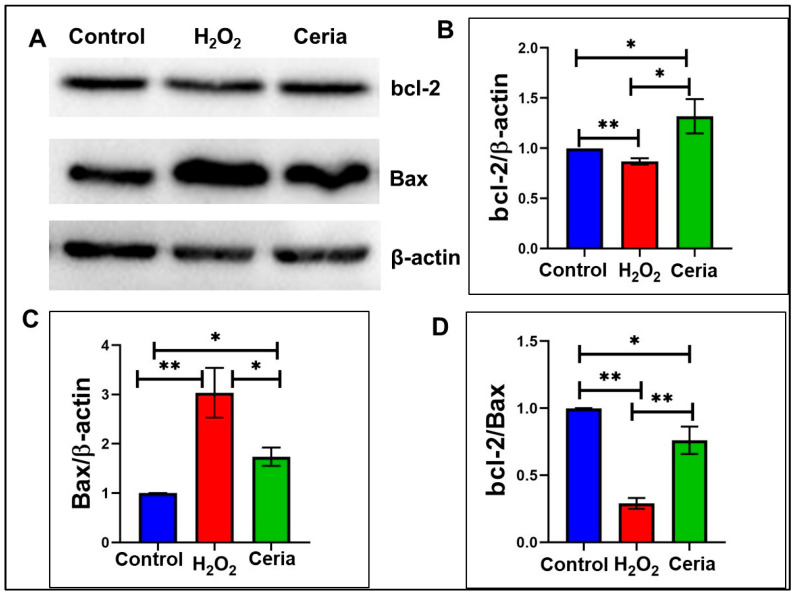
The impact of nanoceria on the expression of proteins related to apoptosis in HT22 cells under oxidative stress caused by H_2_O_2_. (**A**) Immunoblots for bcl-2 and Bax; (**B**) expression levels of bcl-2. (**C**) bax expression levels; and (**D**) bcl-2 to Bax ratio. Statistical significance was determined with * when *p* < 0.05, ** when *p* < 0.01 based on a sample size of n = 3.

**Figure 6 ijms-25-13281-f006:**
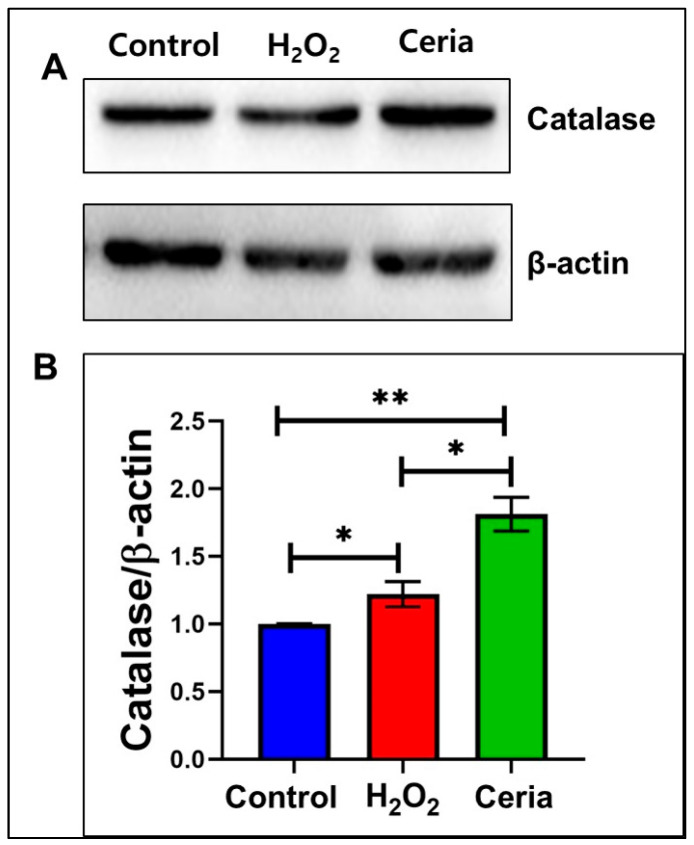
How nanoceria influences catalase levels in HT22 cells under oxidative stress induced by H_2_O_2_. (**A**) Immunoblot images for catalase; (**B**) quantified catalase expression levels. Statistical significance was determined with * when *p* < 0.05 and ** when *p* < 0.01 based on a sample size of n = 3.

**Figure 7 ijms-25-13281-f007:**
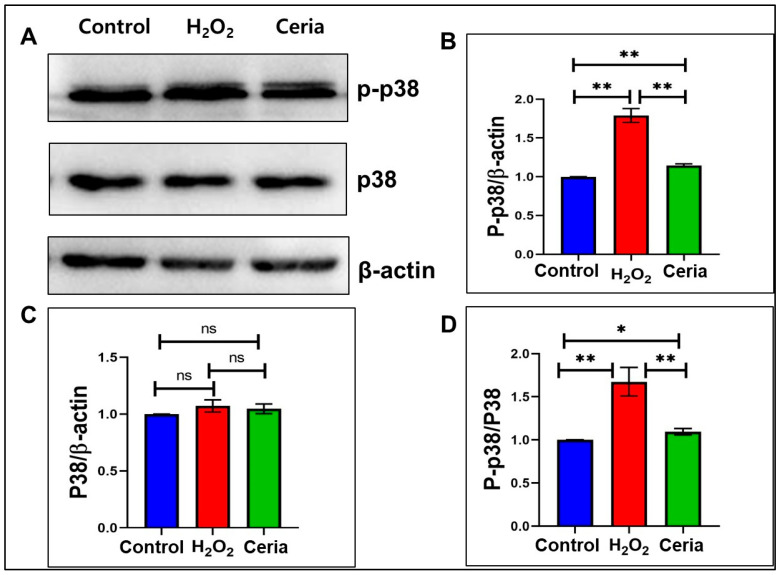
The effect of nanoceria on p-p38 and p38 MAPK expression in HT22 cells subjected to H_2_O_2_-induced oxidative stress. (**A**) Immunoblots for p-p38 and p38; (**B**) relative p-p38 expression; (**C**) relative p38 expression; and (**D**) p-p38 to p38 ratio. * *p* < 0.05, ** *p* < 0.01, and ns indicates not significant (n = 3).

**Figure 8 ijms-25-13281-f008:**
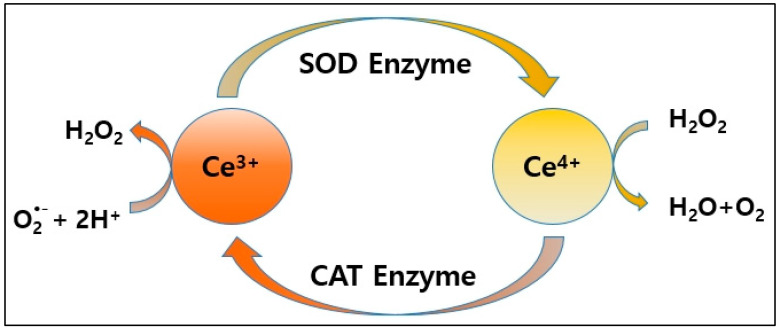
Nanoceria exerts antioxidant effects by scavenging free radicals by mimicking the activities of SOD and CAT [47].

**Table 1 ijms-25-13281-t001:** The antibodies used in this study are listed below.

Antibody	Company	Item Number	Attributes	Dilution Ratio
bcl-2	Santa Cruz Biotechnology	sc-7382	Mouse	1:500
Bax	Santa Cruz Biotechnology	sc-23959	Mouse	1:500
catalase	Santa Cruz Biotechnology	sc-271803	Mouse	1:1000
p-p38	Cell Signaling	4511	Rabbit	1:1000
p38	Cell Signaling	9212	Rabbit	1:1000
β-actin	Cell Signaling	Sc-47778	Mouse	1:1000

## Data Availability

The datasets generated and analyzed during this study are available from the corresponding author upon reasonable request.

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
