# Peer review of "Exploring the Antioxidant Mechanisms of Nanoceria in Protecting HT22 Cells from Oxidative Stress"

_ijms, 2024, doi:10.3390/ijms252413281_

Round 1
Reviewer 1 Report
Comments and Suggestions for Authors
This research paper by Dong et al. examined the neuroprotective function of nanoceria, a nanoenzyme exhibiting antioxidant properties, in safeguarding HT22 cells against oxidative stress caused by hydrogen peroxide (H₂O₂). The study discovered that Nanoceria elevates the expression of the anti-apoptotic protein bcl-2 while diminishing the expression of the pro-apoptotic protein Bax, hence augmenting the bcl-2/Bax ratio, which substantiates its anti-apoptotic impact. It is also shown to enhance catalase expression and inhibit the p38 MAPK signaling pathway, demonstrating its function in protecting HT22 cells from oxidative stress damage. This research offers empirical evidence for the prospective uses of nanoceria in skin anti-aging and the prevention and treatment of various oxidative stress-related ailments. The researchers employed different doses of H₂O₂ to generate oxidative stress in HT22 cells and evaluated cell survival. They evaluated several concentrations of nanoceria to ascertain its effective dosage for cellular protection. Western Blot analysis was performed to evaluate the expression of critical apoptosis-related proteins and catalase. The study's results indicated that Nanoceria markedly enhanced cell viability in oxidative stress situations. The results indicate that nanoceria possesses considerable promise as an antioxidant therapy for safeguarding cells against oxidative damage and facilitating tissue repair, with applications in dermatological and medical interventions for disorders associated with oxidative stress. I have the following suggestions related to the manuscript.
1. The authors claimed impacts on skin-aging in several sentences which is a hypothesis without any true evidence in animals or humans. The research was conducted in vitro, which limits the ability to directly translate the findings to in vivo.
2. The research concentrated on HT22 cells, a particular mouse hippocampal neuronal cell line. Outcomes may differ among various cell types or within human cells. What’s the reason behind choosing this specific cell line?
3. The investigation employed defined concentrations of nanoceria and hydrogen peroxide. Various concentrations or quantities may produce disparate outcomes, and the ideal therapeutic dose for human application remains undetermined or were not fully explored.
4. The study primarily examined short-term effects, neglecting long-term consequences. The long-term effects and potential toxicity of nanoceria require additional research and I suggest the authors to cite relevant literature on toxicology evaluation in vivo at the doses that the authors have tested if there is any published evidence.
5. The study offered insights into the protective mechanisms of nanoceria but did not completely clarify all the molecular processes involved. Further comprehensive mechanistic investigations are necessary in the discussion section.
6. The behavior of nanoceria in intricate biological systems and its interactions with many environmental elements were not well examined. Can the authors briefly discuss it in the discussion?
Author Response
Reviewer #1
- Q:The authors claimed impacts on skin-aging in several sentences which is a hypothesis without any true evidence in animals or humans. The research was conducted in vitro, which limits the ability to directly translate the findings to in vivo.
A: We agree with the reviewers' comments and plan to address the current research topic through the following three steps:
- In Vitro Antioxidant Experiments: At present, we are conducting in vitro antioxidant experiments. Based on the existing results, we can only preliminarily hypothesize that nanoceriamay have potential anti-photoaging effects on UV-exposed skin in real-life scenarios.
- 2. Antioxidant Studies in Animal Models: In the second step, we plan to use mice or rats as experimental models to verify the anti-photoaging effects of nanoceria under actual UV exposure conditions. This step will focus on two key aspects:
1) Verifying the physical shielding effect of nanoceria against UV radiation.
2) Assessing their effectiveness in preventing UV-induced skin damage.
- Translational Research for Practical Applications: In the third step, we will integrate data from both in vitro and in vivo experiments to explore the practical applications of nanoceria. This translational process will require collaboration with experienced companies to facilitate the successful application of the research findings.
Through these three steps, we aim to comprehensively validate the anti-photoaging effects of nanoceria and transform their potential into practical applications.
- Q: The research concentrated on HT22 cells, a particular mouse hippocampal neuronal cell line. Outcomes may differ among various cell types or within human cells. What’s the reason behind choosing this specific cell line?
A: From an anatomical perspective, the junction between the epidermis and dermis is rich in nerve endings. UVB radiation, which is primarily responsible for sunburn, penetrates to this depth, stimulating and damaging these nerve endings, thereby causing acute inflammatory reactions such as redness, swelling, heat, and pain. In our experiment, we selected a hydrogen peroxide concentration of 500 μM as the optimal oxidative stress condition, as it maintains cell viability at 70%–80%. Previously, we tested the optimal oxidative stress concentration using skin fibroblasts, which was found to be 1 mM. This suggests that nerve cells are more sensitive to oxidative stress than fibroblasts. Based on this finding, we chose more oxidative stress-sensitive cells to better protect the skin from UV damage. This choice is also supported by dermatological anatomy.
From a translational perspective, using human cells is undoubtedly more appropriate, and in this regard, I agree with the reviewers' comments. However, when using human cells in animal experiments, the use of immunosuppressants presents a complex challenge. Additionally, experimental data obtained through this approach tend to exhibit greater variability. Experienced researchers in translational studies are well-equipped to bridge the gap between animal experiments and human applications, so this process should not pose significant issues, and the specifics need not be elaborated here. Alternatively, using animal cells for in vitro experiments and the same species for in vivo studies ensures consistency across the experimental stages, leading to more reliable data.
In conclusion, our research design not only accounts for cell sensitivity and experimental consistency but also balances the rationale and feasibility of progressing from basic research to translational applications.
- Q: The investigation employed defined concentrations of nanoceria and hydrogen peroxide. Various concentrations or quantities may produce disparate outcomes, and the ideal therapeutic dose for human application remains undetermined or were not fully explored.
A: We fully agree with the reviewers' comments. In our previous studies, we have confirmed that the dosages of nanoceria and hydrogen peroxide vary depending on the cell types used. Regarding the issues raised in the current review, some aspects have already been addressed in our responses to the previous two questions. Below, we provide additional explanations:
First, this study represents the initial phase of the entire research project, focusing on validating the antioxidant properties of nanoceria through in vitro experiments and exploring its underlying antioxidant mechanisms. In the second phase, which involves animal experiments, the dosage of nanoceria will be determined based on prior experience, typically set at approximately ten times the in vitro dosage. Of course, adjustments may be made during the actual experiments as needed.
For the third phase, translational research, we plan to collaborate closely with experts specializing in translational studies. They will use the data obtained from the first two phases as a reference, combined with their extensive research experience, to preliminarily determine the appropriate dosage for human use. During the translational research process, the dosage will be gradually optimized to identify the final optimal dosage, ensuring that the research outcomes are both safe and effective for practical applications.
Through this stepwise research design, we aim to build a scientific bridge between experimental data and practical applications, thereby facilitating the successful translation of research findings into real-world benefits and maximizing their societal value.
- Q: The study primarily examined short-term effects, neglecting long-term consequences. The long-term effects and potential toxicity of nanoceria require additional research and I suggest the authors to cite relevant literature on toxicology evaluation in vivo at the doses that the authors have tested if there is any published evidence.
A: We fully agree with the reviewer’s opinion that the toxicological issues of nanomaterials are indeed complex and cannot be comprehensively explained within a short response. Before addressing the specific questions raised by the reviewer, we first need to clarify the main pathways through which foreign substances, including nanomaterials, enter the human body. These pathways generally include the skin, respiratory tract, digestive tract, and bloodstream. The reactions triggered within the body vary significantly depending on both the amount of foreign material and the duration of exposure.
We have reviewed a substantial amount of literature, but due to differences in research focus and experimental methods, the findings sometimes lead to entirely opposite conclusions. Regarding the “short-term effects and long-term outcomes” mentioned by the reviewer, our study is based on in vitro experiments using cells, which limits our observations to short-term effects. Animal studies, on the other hand, would require several months or even longer periods for observation to draw conclusions about long-term effects.
To address this issue, we have added relevant discussions in the revised manuscript by referencing similar studies and providing comparative analyses. These modifications have been highlighted in blue font on page 4~5 of the revised manuscript for your review.
- Q: The study offered insights into the protective mechanisms of nanoceria but did not completely clarify all the molecular processes involved. Further comprehensive mechanistic investigations are necessary in the discussion section.
A: We have addressed the reviewers' suggestions by adding the requested supplementary content to the discussion section and emphasized it using blue font.
- Q: The behavior of nanoceria in intricate biological systems and its interactions with many environmental elements were not well examined. Can the authors briefly discuss it in the discussion?
A: We agree with the reviewers' comments and have added the suggested content to the discussion section, emphasizing it using blue font.

Reviewer 2 Report
Comments and Suggestions for Authors
The submitted manuscript presents the results of original study that investigates the neuroprotective properties of nanoceria against oxidative stress caused by hydrogen peroxide (H₂O₂) in HT22 cells. The manuscript is well written and nicely presented, also within the scope of the Special Issue of IJMS that it was submitted to. The similarity level is acceptable, the quality of figures is appropriate. The amount of newly presented data could have been increased, but it is acceptable. However, some parts of this manuscript need clarification and revision. I’ve pointed out the direct issues below.
Line 12, please replace „leads” with „leading”
Line 42, well, not only from Chinese…. I suggest to remove this adjective
Lines 44-45, the definition of nanozyme should be presented here
Line 48, at this point it must be stated what nanoceria really is, from the chemical/structural point of view
In the introduction the authors should inform the readers that toxicological reports on cerium compounds have noted their cytotoxicity and contributions to pulmonary interstitial fibrosis in workers.
Line 91, it should be “30 minutes”
Line 322, what do you mean by “< 25nm”?
Line 366, what exactly statistical test has been used? Tukey?
Line 99, justify, in one sentence, why not 750 μM concentration has been chosen, just to “be secured”?
Figure 2, what were the p-values? The “80” seems to be more altered, in comparison to control, than i.e. “20”, which received “*”. I suggest to repeat the statistical analysis and provide the exact p-values.
Line 133, figure caption is incorrect, I suppose it should be “†p<0.05” not “††p<0.05”
Line 262, equation (1) is invalid, as the charges don’t balance, (+5 left / +4 right )
Line 268, the role of SOD and CAT should be mentioned in the introduction as well
A “Conclusions” section is missing. Adding it would be beneficiary.
The Authors should consider the possible metabolism and elimination of the nanoceria, in terms of potential bioaccumulation, which can be dangerous.
Author Response
Reviewer #2
- Q:Line 12, please replace „leads” with „leading”
A: Please refer to line 12, where I have changed "leads" to "leading" and highlighted it in red for emphasis.
- Q: Line 42, well, not only from Chinese…. I suggest to remove this adjective
A: Please refer to line 42, where the deletion has been made according to the reviewer’s suggestion.
- Q:Lines 44-45, the definition of nanozyme should be presented here
A: Please refer to lines 45-46, where the definition of nanozymes has been supplemented and highlighted in red.
- Q:Line 48, at this point it must be stated what nanoceria really is, from the chemical/structural point of view
A: Please refer to line 51-52, where the description of nanoceria has been supplemented and highlighted in red.
- Q:In the introduction the authors should inform the readers that toxicological reports on cerium compounds have noted their cytotoxicity and contributions to pulmonary interstitial fibrosis in workers.
A: We agree with the reviewer's comments and have highlighted the toxicological issues related to nanoceria in the introduction using red font.
- Q:Line 91, it should be “30 minutes”
A: The changes have been made in line 117 according to the reviewer's suggestion.
- 7. Q: Line 322, what do you mean by “< 25nm”?
A: As shown in the figure, the antioxidant activity of nanoceria is closely related to its particle size, which is why we have highlighted it specifically.
- Q:Line 366, what exactly statistical test has been used? Tukey?
A: The post hoc Tukey's test was used, and we have made a note of this in the statistics section, highlighted in red.
- Q:Line 99, justify, in one sentence, why not 750 μM concentration has been chosen, just to “be secured”?
A: Based on experimental experience, to observe the antioxidant effects of nanoceria within the specified time, if the cells experience stronger oxidative stress, their viability will significantly decrease, indicating more severe damage to the cells, which would result in not obtaining the expected results.
- Q:Figure 2, what were the p-values? The “80” seems to be more altered, in comparison to control, than i.e. “20”, which received “*”. I suggest to repeat the statistical analysis and provide the exact p-values.
A: Following the reviewer's suggestion, we conducted a reanalysis of the statistics and corrected Figure 2 based on the new results. Additionally, we have added and revised the analysis section of the manuscript, highlighted in red.
- Q:Line 133, figure caption is incorrect, I suppose it should be “†p<0.05” not “††p<0.05”
A: The corrections have been made according to the reviewer's suggestion, and red font has been used to highlight them in line 176.
- Q:Line 262, equation (1) is invalid, as the charges don’t balance, (+5 left / +4 right )
A: We agree with the reviewer's comment and have adjusted the charge balance in line 307.
- Q:Line 268, the role of SOD and CAT should be mentioned in the introduction as well
A: Based on the reviewers' suggestions, we have moved the roles of SOD and CAT to the introduction section and highlighted the changes in red font for clarity.
- Q:A “Conclusions” section is missing. Adding it would be beneficiary.
A: Based on the reviewer’s' suggestions, we have moved the "In summary" from the Discussion section to the Conclusions section, with some additional content added for better clarity and completeness. The changes have been highlighted in red font for emphasis.
- Q:The Authors should consider the possible metabolism and elimination of the nanoceria, in terms of potential bioaccumulation, which can be dangerous.
A: Based on the reviewers' suggestions, we have added content in the Discussion section regarding the possible metabolism and elimination of nanoceria, including potential bioaccumulation issues. The new content has been highlighted in red font.

Round 2
Reviewer 2 Report
Comments and Suggestions for Authors
The Authors have revised and improved their work. This version can be accepted for publication.